# Transient Neutropenia in Immunocompetent Infants with Respiratory Syncytial Virus Infection

**DOI:** 10.3390/v13020301

**Published:** 2021-02-15

**Authors:** Tatsuya Korematsu, Hiroshi Koga

**Affiliations:** Department of Pediatrics, National Hospital Organization Beppu Medical Center, 1473 Oaza-Uchikamado, Beppu, Oita 874-0011, Japan; tatsuya.korematsu@gmail.com

**Keywords:** host microbial interactions, leukopenia, neutrophils, respiratory tract infections, severity of illness index

## Abstract

The incidence of neutropenia and the association between neutropenia and severity of respiratory symptoms among infants with respiratory syncytial virus (RSV) infections remain to be elucidated. This single-center, retrospective study included immunocompetent infants (<10 months old) with laboratory-confirmed RSV infection admitted to our center between January 2012 and December 2019. Incidence of neutropenia (<1.0 × 10^9^/L) within 10 days of onset and risk factors associated with subsequent neutropenia were evaluated. Among the 292 infants with RSV infection, including 232 (79%) with mild infection, neutropenia was observed in 31 (11%), with severe neutropenia (<0.5 × 10^9^/L) in 3 (1.0%). No neutropenic infants developed serious infection or hematological disorder. Infants without neutropenia showed age <3 months at onset in 34%, C-reactive protein level <1.0 mg/L in 27%, and nasopharyngeal microbiota composition with any of *Moraxella catarrhalis*, *Streptococcus pneumoniae*, or *Haemophilus influenzae* in 63%. In comparison, infants with neutropenia showed age <3 months at onset in 74% (relative risk [RR] 2.15; 95% confidence interval [CI] 1.65–2.81), C-reactive protein level <1.0 mg/L in 55% (RR 2.02; 95% CI 1.38–2.94), and microbiota including *Moraxella catarrhalis*, *Streptococcus pneumoniae*, or *Haemophilus influenzae* in 15% (RR 0.24; 95% CI 0.10–0.61). Multiple logistic regression analyses showed that younger age at onset and absence of that nasopharyngeal microbiota profile were associated with development of neutropenia. In conclusion, age and airway microbiota are considered as risk factors for the development of transient neutropenia among infants with RSV infection. However, the neutropenia seems not to develop serious infection or hematological disorder.

## 1. Introduction

Respiratory syncytial virus (RSV) continues to cause significant morbidity and mortality among children and the elderly worldwide [1]. Approximately 33 million annual episodes of RSV-associated lower respiratory infection occur among children <5 years, resulting in 3 million hospital admissions and 60,000 in-hospital deaths [2]. Nearly half of these admissions and deaths occur in infants <6 months [2].

To reduce the global burden of RSV infection, RSV vaccines and antivirals have been developed and are currently in clinical trials [3]. Lumicitabine (ALS-008176 or JNJ-64041575), an oral nucleoside analogue against RSV polymerase, has shown successful results in phase I and II trials in adults [4]. However, adverse effects of neutropenia were observed in a phase II trial of infants <3 years old, and the trial was terminated [5]. To clarify the causal relationship, additional nonclinical study (monkey study) was undertaken and showed potential clastogenic effects (data not shown). In clinical practice, neutropenia subsequent to RSV infection, particularly persistent neutropenia, influences the decision to perform a diagnostic work-up for serious infectious and hematological diseases.

Transient neutropenia can be observed in immunocompetent infants following viral infections, including RSV infection [6,7,8]. However, the cumulative incidence of neutropenia in RSV-infected infants remains to be elucidated. The aim of this study was to determine the incidence of neutropenia among infants with RSV infection and to investigate risk factors for neutropenia following RSV infection. Our results provide insights into clinical practice for RSV-infected infants and pathways to further clinical trials of RSV antivirals.

## 2. Methods

### 2.1. Study Design and Setting

This was a retrospective observational study of infants admitted to a single pediatric center in Japan for RSV infection. Beppu Medical Center is a pediatric center for the 100,000 inhabitants of the city of Beppu, in the southeast of Japan. In this city, all pediatric patients requiring emergency care or hospital admission are transported to our center. This study protocol, based on the 2007 version of the Strengthening the Reporting of Observational Studies in Epidemiology (STROBE) statement [9], was approved by the ethics committee of Beppu Medical Center. Passive informed consent was obtained from the parents of participants through the hospital website and bulletin board.

### 2.2. Study Participants

In this study, we used a global respiratory severity score (GRSS) [10], a clinical severity score validated for full-term infants <10 months of age with RSV infection. Thus, we reviewed the information of full-term infants who were <10 months old at the time of hospitalization for RSV infection at our center between January 2012 and December 2019. RSV infection was diagnosed based on rapid immunochromatographic antigen testing of nasopharyngeal swabs or aspirates for RSV. The date of consecutive fever (>37.5 °C) or respiratory symptom occurrence was considered as the first day of RSV infection. Infants with at least one complete blood count performed within 10 days of illness were included. Exclusion criteria were as follows: neutropenia diagnosed before onset of RSV infection, other preceding hematological disorder, malignancy, immune disorder, inborn errors of metabolism, administration of drugs associated with myelosuppression, administration of palivizumab within the preceding month, or enrollment in a clinical trial for RSV antiviral agents.

### 2.3. Data Collection and Definitions

Information about baseline characteristics, physical findings, laboratory data, and treatments during the course of RSV infection were also obtained from the electronic medical records of eligible infants. Clinical severity of RSV infection was assessed on admission using a GRSS. GRSS is calculated using nine variables: general appearance; wheezing; rales/rhonchi; chest wall retractions; cyanosis; lethargy; poor air movement; worst room air SaO_2_; and maximum respiratory rate [10]. GRSS is reportedly associated with length of hospital stay and to affect the CD8^+^ T-cell profile in infants with RSV infection [10,11]. Neutropenia in infants <1 year old is defined as an absolute neutrophil count (ANC) <1.0 × 10^9^/L (severe: <0.5 × 10^9^/L) [8]. The nutritional condition of infants was assessed by the z-score of weight-for-length, as calculated using the Anthro Survey Analyser by the World Health Organization [12]. Abnormal values for body temperature (>38.5 °C), heart rate (>180 beats/min), and respiratory rate (>34 breaths/min) were defined according to systemic inflammatory response syndrome criteria for infants [13].

Analysis of microbial genomic sequences from nasopharyngeal aspirates in infants with RSV infection demonstrated *Streptococcus*-, *Moraxella*-, and *Haemophilus*-dominant profiles as the three major microbial profiles, and these nasopharyngeal microbiota profiles were associated with the clearance of RSV [14]. Based on this report, the microbiota profile was evaluated from bacterial cultures of nasopharyngeal aspirate in this study.

### 2.4. Outcome Measures

The primary outcome measure was neutropenia in infants with RSV infection within 10 days of onset. Secondary outcome measures were the potential risk factors including GRSS associated with the occurrence of neutropenia.

### 2.5. Follow-Up Assessment

Infants showing neutropenia during hospitalization for RSV infection were evaluated for clinical signs of underlying serious infectious and hematological disorder, including pancytopenia and peripheral blood blast cells. When no clinical signs suggested underlying serious infectious or hematological diseases, infants with neutropenia were discharged after recovery from RSV infection. The parents of infants with neutropenia were sent a questionnaire in May 2020 to identify the recovery of neutropenia and subsequent onset of any hematological disorder.

### 2.6. Statistical Analysis

Descriptive statistics were used to summarize basic and clinical characteristics, as percentages. Bivariate associations were assessed using Pearson’s χ^2^ test or Fisher’s exact test. Wilcoxon rank-sum test was used to compare continuous variables. Patient characteristics, GRSS, laboratory data, and treatment during the course of RSV infection were compared between infants with and without neutropenia. Logistic regression analysis was used to identify independent risk factors for neutropenia. The multivariate model was adjusted for GRSS, potential confounders on univariate analysis (*p* < 0.05), and factors previously reported as relevant to neutropenia, such as age and sex [8,15].

JMP version 14 software (SAS Institute Inc., Cary, NC, USA) was used for all statistical analyses. Values of *p* < 0.05 were considered significant. Measures of effect are presented as the risk ratio or odds ratio with 95% confidence interval (CI).

## 3. Results

### 3.1. Basic Characteristics

During the study period, a total of 308 infants <10 months old were admitted to our hospital because of RSV infection. RSV infection was confirmed by rapid immunochromatographic antigen testing in all infants. Among these infants, complete blood counts were performed within 10 days of onset in the 298 infants who met the inclusion criteria, then six infants were excluded (due to palivizumab administration within 1 month in five infants; and participation in a clinical trial of RSV antiviral in one infant). A total of 292 infants were subsequently considered eligible for this study. Among the 292 infants, 60 (21%) had GRSS scores of >3.5 [10], a threshold for predicting hospitalization, and 232 (79%) had less severe GRSS scores of ≤3.5. All 292 infants underwent complete blood counts and differential evaluations at admission and 70 (24%) underwent two times or more blood tests within 10 days of onset.

The ethnicity of the 292 eligible infants comprised 290 (99%) Asian (286 Japanese, 2 Korean, 1 Chinese, and 1 Uzbek), 2 (0.7%) white, and no black. A seasonal trend in admissions of RSV-infected infants was observed as an increase from September to December in 151 infants (52%) and a decrease from April to July in 32 (11%).

Among the 292 infants with RSV infection, median age at admission was 4.2 months (interquartile range [IQR] 2.1–7.3 months). Neutropenia with ANC <1.0 × 10^9^/L was identified in 31 (11%) of these 292 infants, including severe neutropenia with ANC <0.5 × 10^9^/L in 3 (1.0%). In addition, neutropenia with ANC <1.0 × 10^9^/L was observed in 32 (8.7%) of the 368 complete blood counts performed within 10 days of illness. The incidence of neutropenia according to the day of illness in a series of complete blood counts is illustrated in Figure 1. Neutropenia was observed in approximately 5–15% of blood samples from days 2 through 9 of illness in RSV-infected infants.

Bacterial cultures of nasopharyngeal aspirates were collected from 234 (81%) of 292 infants (26 with neutropenia, 208 without neutropenia). Blood cultures collected from 26 infants were all negative. During hospitalization, apnea was observed in five infants (1.7%). In total, 289 infants (99%) were treated with inhaled bronchodilator or nebulized hypertonic saline, 282 (97%) with intravenous fluid administration, 88 (30%) with oxygen therapy, 58 (20%) with antibiotic administration, 14 (4.8%) with corticosteroid administration, nine (3.1%) with high-flow nasal cannula, and one (0.3%) with endotracheal intubation. None of the infants developed subsequent leukemia or other bone marrow disorder, or died during the course of RSV infection.

### 3.2. Potential Risk Factors for Neutropenia

Among the 292 infants, a comparison of clinical manifestations between those with and without neutropenia is summarized in Table 1. Comorbidities were observed in 10 infants without neutropenia, including congenital heart defects in four, central nervous system abnormalities in three, cleft lip and palate in one, osteogenesis imperfecta in one, and anhidrotic ectodermal dysplasia in one. Younger age at onset, lower C-reactive protein level, and less frequent identification of any of *Moraxella catarrhalis*, *Streptococcus pneumoniae*, or *Haemophilus influenzae* in nasopharyngeal aspirates were observed in infants with neutropenia, compared to those without neutropenia. Median ANC and C-reactive protein level on admission were higher in the 135 infants with any of these three bacteria identified from nasopharyngeal aspirates (4.0 × 10^9^/L and 8.1 mg/L) than in the 99 infants without such a microbiota profile (2.0 × 10^9^/L and 0.9 mg/L, *p* < 0.0001 and *p* < 0.0001, respectively). Age at onset of RSV infection <3 months represented an absolute risk increase of 40% for subsequent neutropenia (number needed to diagnose = 2.5 infants). No significant difference in corticosteroid use and severity of RSV infection as evaluated by GRSS was apparent between groups.

Multiple logistic regression analysis was performed to evaluate associations between neutropenia and relevant factors such as sex, age at onset, C-reactive protein level, nasopharyngeal microbiota profile, and GRSS (Table 2). Neutropenia within 10 days of onset in RSV-infected infants was associated with younger age at onset and identification of any of the three bacterial species mentioned above in cultures of nasopharyngeal aspirate, but was no longer associated with sex, C-reactive protein level, or GRSS.

### 3.3. Follow-up Assessment

For the 31 neutropenic infants, 21 questionnaires (68%) were returned. Median follow-up period was 3.6 years (IQR 1.8–5.0 years). None of the 21 infants developed hematological disorder or other serious diseases after hospital discharge. Complete blood count was performed in 19 (90%) of the 21 infants after discharge, and all 19 infants had recovered (ANC >1.5 × 10^9^/L) from neutropenia.

## 4. Discussion

This study elucidated the incidence and the risk factors of transient neutropenia in infants with RSV infection. Among immunocompetent infants with RSV infection, 11% of the infants showed the transient neutropenia (ANC <1.0 × 10^9^/L), and the transient neutropenia demonstrated a correlation with age at onset and nasopharyngeal microbiota profile (presence of *M. catarrhalis*, *S. pneumoniae*, or *H. influenzae*). Among RSV-infected infants with transient neutropenia, no subsequent serious secondary infections, bone marrow disorders, or deaths were observed. Our results suggest that comorbid neutropenia is a transient phenomenon without developing severe infection or hematologic malignancy among infants with RSV infection.

### 4.1. RSV Infection and Neutropenia

The estimated incidence of transient neutropenia among RSV-infected infants is 11% in this study, which raises concerns about potential serious bacterial infection or bone marrow disorder, particularly in prolonged neutropenia. The incidence of serious bacterial infection has been reported as 3.7–4.2% among young febrile infants with viral infections [16,17], 2.1–8.5% among immunocompetent children with febrile neutropenia [7,18], and 1.1–7.0% among young infants with RSV infection [17,19,20]. These risks are uncommon, but non-negligible. Based on these results, targeted antibiotic treatment may be needed in RSV-infected infants with febrile neutropenia. Prophylactic antibiotic administration may provide clinical benefit for RSV-infected infants with febrile neutropenia [18], especially in young or ill-appearing infants [19,20]. Bone marrow disorders, including leukemia or myelodysplastic syndrome, were found in 1.5–2.5% of previously healthy, infected children with febrile neutropenia, and all of them showed chronic neutropenia >2 months [6,21]. Given the safety concerns, a diagnostic work-up of underlying serious etiologies is necessary for RSV-infected infants with febrile or prolonged neutropenia. Prophylactic antibiotic use may then be withheld from RSV-infected infants with afebrile neutropenia.

### 4.2. Age-Related Prevalence of Neutropenia

Normal values for ANC vary by age and ethnicity [8]. Except for the first two weeks after birth, 1.0 × 10^9^/L is used as the lower limit of normal for ANC during the first year of life, with 1.5 × 10^9^/L thereafter [8]. Benign ethnic neutropenia has been described in black individuals [8,15], but no such individuals were present in this study. In our study, the cumulative incidence of neutropenia (<1.0 × 10^9^/L) was 11% among RSV-infected infants, higher than <5% among infants from the general population [8]. Febrile neutropenia was reported as a serious adverse event in the clinical trial of lumicitabine, a nucleoside analogue against RSV polymerase, in RSV-infected children <3 years old [5]. However, neutropenic complications have not been reported in adult clinical trials of lumicitabine or other nucleoside polymerase inhibitors, such as PC786 or EDP-938 [22,23].

### 4.3. Inflammatory Effect of Airway Microbiota in RSV Infection

In the present study, transient neutropenia occurring in infants with RSV infection showed an association with nasopharyngeal microbiota, including *M. catarrhalis*, *S. pneumoniae*, or *H. influenzae*. Both ANC and C-reactive protein levels were higher in RSV-infected infants with such microbiota than in those without. These results indicate that nasopharyngeal microbiota profile can potentially induce systemic inflammation and neutrophil recruitment. The effects of the airway microbiota profile on inflammatory response and on the severity of RSV infection have been studied. Nasopharyngeal microbiota comprising *H. influenzae* or *Streptococcus* have shown correlations with overexpression of inflammation-related genes and the need for hospitalization in RSV-infected children <2 years old [24]. In addition, nasopharyngeal microbiota with *H. influenzae* was associated with delayed clearance of RSV and the need for intensive care and for prolonged hospitalization in RSV-infected infants [14,25]. Airway microbiota, including *Moraxella* or *Haemophilus,* evoked the release of cytokines and chemokines among non-infected healthy neonates and among RSV-infected infants, contributing to the development of recurrent wheezing until 3 years of age [26,27].

### 4.4. Potential Role of Airway Microbiota Composition in Neutropenic Phenomenon

Our results demonstrated that nasopharyngeal microbiota composition without *M. catarrhalis*, *S. pneumoniae*, or *H. influenzae* correlated with increased incidence of transient neutropenia and was not correlated with the severity of RSV infection in infants. This suggests that decreased circulating neutrophil counts do not necessarily imply immunosuppression. RSV can be detected in peripheral blood in children, and infects bone marrow stromal cells in children and adults [28,29]. RSV infection in infants also enhances the release of hematopoietic cytokines, inflammatory cytokines, and proinflammatory chemokines in nasal lavage fluids [30]. The release of these proinflammatory cytokines and chemokines induced by RSV infection has an effect on the recruitment and the migration of neutrophils to inflammatory sites. Decreased circulating neutrophils in the peripheral blood of infants with RSV infection are therefore considered to result from facilitated transmigration and extravasation [31], in addition to the effect of transient myelosuppression or neutrophil apoptosis. This postulated mechanism is consistent with the infrequent occurrence of serious bacterial infections among immunocompetent children with febrile neutropenia [7,18]. Transient neutropenia in RSV-infected infants has been reported to persist for approximately 30 days [6]. Further research is required to reveal the pathogenesis of transient neutropenia in infants with RSV infection.

### 4.5. Limitations

This type of retrospective observational study has inherent limitations. First, the timing and the frequency of collecting blood samples within 10 days of illness were not the same for all participants. This may underestimate the incidence of benign neutropenia in RSV-infected infants. Second, 32% of patients did not complete follow-up assessment in this study, and missing data were not included in the analyses. We could not confirm the recovery of neutropenia in all RSV-infected infants with neutropenia. Duration of neutropenia was thus not analyzed in this study. Missed diagnoses of serious infection, bone marrow disorder, and malignancy following neutropenia in RSV-infected infants were presumed to be rare, because all children requiring emergency care or hospital admission in this city were transported to our center. Third, subtypes of RSV were not identified in the present study. The subtype of RSV has an effect on disease severity, and possibly on ANC, through the enhanced release of proinflammatory cytokines and chemokines [32]. Finally, 80% of infants included in this study showed mild severity of RSV infection (GRSS ≤3.5) [10]. Our study results have limited generalizability to infants with severe RSV infection or with non-Asian ethnicity.

## 5. Conclusions

Among infants hospitalized due to predominantly mild RSV infection, 11% showed transient neutropenia <1.0 × 10^9^/L within 10 days of onset. However, no development of serious disease was observed in the neutropenic infants. Age at onset and nasopharyngeal microbiota profile were associated with the development of transient neutropenia. These risk factors for benign neutropenia in RSV-infected infants should thus be considered in clinical practice and clinical trials for RSV antivirals. Further investigation is needed to determine the pathogenesis and the optimal management of transient neutropenia among children with RSV infection.

## Figures and Tables

**Figure 1 viruses-13-00301-f001:**
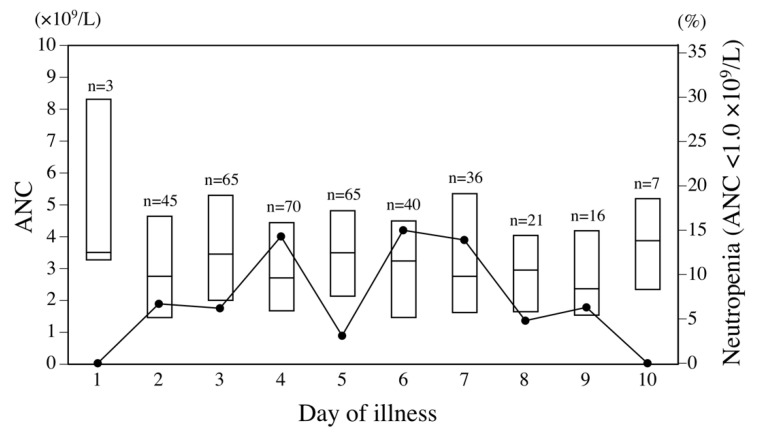
Distribution of absolute neutrophil count (ANC) and proportion of neutropenia <1.0 × 10^9^/L in respiratory syncytial virus (RSV)-infected infants according to day of illness. Each box bounds the interquartile range of ANC, divided by the median. Solid line denotes the proportion of neutropenia.

**Table 1 viruses-13-00301-t001:** Comparison of clinical manifestations between RSV-infected infants with and without neutropenia <1.0 × 10^9^/L.

	Neutropenic Infants (*n* = 31)	Non-neutropenic Infants (*n* = 261)	Relative Risk (95%CI)	*p* Value
**Basic Characteristics**				
Female infant	18 (58)	113 (43)	1.34 (0.96–1.87)	0.12
Age < 3 months at onset	23 (74)	90 (34)	2.15 (1.65–2.81)	<0.0001 *
Weight-for-length z-score < −2	0 (0)	15 (5.8)	not applicable	0.38
Admission before day 5 of illness	22 (71)	156 (60)	1.19 (0.93–1.52)	0.23
**Vital Signs and Oxygen Saturation on Admission**				
Axial temperature > 38.5 °C	2 (6.5)	39 (15)	0.43 (0.11–1.70)	0.28
Heart rate > 180 beats/min	0 (0)	18 (6.9)	not applicable	0.23
Respiratory rate > 34 breaths/min	25 (81)	185 (71)	1.14 (0.94–1.37)	0.25
Arterial oxygen saturation < 95%	2 (6.5)	37 (14)	0.46 (0.12–1.80)	0.40
**Laboratory Findings on Admission**				
Absolute lymphocyte count < 1.5 × 10^9^/L	0 (0)	2 (0.8)	not applicable	1.0
C-reactive protein < 1.0 mg/L	17 (55)	71 (27)	2.02 (1.38–2.94)	0.0015 *
Consolidation on chest radiography	2 (6.5)	52 (20)	0.32 (0.08–1.26)	0.086
**Culture of Nasopharyngeal Aspirate**				
*Moraxella catarrhalis*	1 (3.9) ^†^	87 (42) ^‡^	0.09 (0.01–0.63)	<0.0001 *
*Streptococcus pneumoniae*	1 (3.9) ^†^	51 (25) ^‡^	0.16 (0.02–1.09)	0.013 *
*Haemophilus influenzae*	2 (7.7) ^†^	45 (22) ^‡^	0.36 (0.09–1.38)	0.12
Any of these 3 bacteria	4 (15) ^†^	131 (63) ^‡^	0.24 (0.10–0.61)	<0.0001 *
**Treatments**				
Antibiotic use during course of RSV infection	5 (16)	57 (22)	0.74 (0.32–1.70)	0.64
Corticosteroid use during course of RSV infection	1 (3.2)	14 (5.4)	0.60 (0.08–4.42)	1.0
Hospital stay >7 days	4 (13)	58 (22)	0.58 (0.23–1.49)	0.35
**Severity of RSV Infectio**				
GRSS on admission > 3.5	8 (26)	52 (20)	1.30 (0.68–2.47)	0.44

CI: confidence interval; GRSS: global respiratory severity score; RSV: respiratory syncytial virus. * *p* < 0.05; ^†^ Denominator of 26 (bacterial cultures were not performed in 5); ^‡^ denominator of 208 (bacterial cultures were not performed in 53).

**Table 2 viruses-13-00301-t002:** Logistic regression analysis for predicting neutropenia <1.0 × 10^9^/L in infants with RSV infection.

	Odds Ratio (95%CI)	*p* Value
Any of the 3 bacteria ^†^ identified in cultures of nasopharyngeal aspirate	0.11 (0.03–0.35)	<0.0001 *
Age at onset of RSV infection, per 1-month increment	0.81 (0.65–0.97)	0.020 *
GRSS on admission, per 1-point increment	0.92 (0.67–1.24)	0.59
C-reactive protein, per 1-mg/L increment	1.02 (0.99–1.05)	0.10
Female infant	1.49 (0.61–3.70)	0.38

CI: confidence interval; GRSS: global respiratory severity score; RSV: respiratory syncytial virus. * *p* < 0.05; ^†^
*Moraxella catarrhalis*, *Streptococcus pneumoniae*, or *Haemophilus influenzae.*

## Data Availability

Not applicable.

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
