# Peer review of "Transient Neutropenia in Immunocompetent Infants with Respiratory Syncytial Virus Infection"

_viruses, 2021, doi:10.3390/v13020301_

Round 1

Reviewer 1 Report

This manuscript by Korematsu and Koga describes the The incidence of neutropenia and the association between neutropenia and severity of 
respiratory symptoms among infants with respiratory syncytial virus. Although the research topic is of interest in the context of the immunobiology RSV infection, in this reviewer’s opinion, data provided by the authors is still preliminary. In addition, the hypotheses and aims of this study are largely unclear. A major revision to the text in the manuscript may significantly improve the manuscript.

Major comment:

1. This manuscript lacks an introduction section, which makes it hard for the readership to understand the rationale of the studies conducted, the hypotheses tested, the aims and importantly, its significance to the field. Without this information, it is difficult to accurately interpret the study findings and conclusions.

Author Response

Responses to the Comments from Reviewer 1

We greatly appreciate your review of the manuscript and the helpful suggestions you provided. In the revised manuscript, red text indicates portions revised according to your comments. Our responses to the issues raised are as follows.

  • This manuscript lacks an introduction section, which makes it hard for the readership to understand the rationale of the studies conducted, the hypotheses tested, the aims and importantly, its significance to the field. Without this information, it is difficult to accurately interpret the study findings and conclusions.

Response: Thank you for your comments. Sorry for the lack of introduction. We failed to move the document of introduction section on the template.

Reviewer 2 Report

This is a retrospective observational study performed in a referral centre evaluating the incidence and the risk factors of neutropenia in infants with RSV infection. There are few reports in the literature for immunocompetent infants with RSV infection, thus results are interesting.   Major comments   1. Title: use “transient” neutropenia 2. Introduction: page1 lines 29-39 the main text is missing 3. Results Page 6, lines 186-190 Please delete (irrelevant text) Page 4 line 147: write “between those with...” Page 4 line 152 and page 5 table 1: text mentions that more frequent identification of any of the three bacteria was observed in infants with neutropenia. On the contrary, table 1 shows that any of these three bacteria was identified in 4/26 (15%) of neutropenia infants as opposed to 131/208 non neutropenic infants. This creates confusion, please check the accuracy of data 4. Discussion: Use the term “transient neutropenia” instead of “neutropenia” to better describe the results Page 6 line 195 please clarify-explanation negative correlation between neutropenia and nasopharyngeal microbiota profile. We use negative correlation for continuous variables. Page 6 line 199, this retrospective study did not assess need for prophylactic antibiotic treatment which was administered in 20% of patients. Therefore this statement should be omitted or discussed with caution Page 6, lines 201-215 make it more concise and discuss your results from the perspective of the related studies Page 6 line 216-227 discuss your results regarding febrile neutropenia Page 7 lines 228-244 check for consistency with page 4 line 152 The fact that C-rp is higher in RSV infected infants with microbiota is not mentioned in the results and appears in the discussion Page 7 lines 245-268 make this paragraph more concise as your study provides no data on cytokines and inflammatory response. You only evaluated ANC. Page7,8 lines 269-283 please include in limitations that 32% of patients did not complete follow up assessment, missing data was not performed.

Author Response

Responses to the Comments from Reviewer 2

We greatly appreciate your review of the manuscript and the helpful suggestions. In the revised manuscript, blue text indicates portions revised in accordance with your comments. Our responses to the issues raised are as follows.

  • Title: use “transient” neutropenia

Response: Title is changed according to your suggestion.

  • Introduction: page1 lines 29-39 the main text is missing

Response: Sorry, introduction is added and margin is deleted. There is no other missing text.

  • Results Page 6, lines 186-190 Please delete (irrelevant text) Page 4 line 147: write “between those with...” Page 4 line 152 and page 5 table 1: text mentions that more frequent identification of any of the three bacteria was observed in infants with neutropenia. On the contrary, table 1 shows that any of these three bacteria was identified in 4/26 (15%) of neutropenia infants as opposed to 131/208 non neutropenic infants. This creates confusion, please check the accuracy of data.

Response: Thank you for careful suggestions. Irrelevant text in Page 6 is deleted. “between with” is corrected to “between those with” in Page 4. In Table 1, “in” is also changed to “between”. In page 5, “more frequent identification” is changed to “less frequent identification”.

  • Discussion: Use the term “transient neutropenia” instead of “neutropenia” to better describe the results Page 6 line 195 please clarify-explanation negative correlation between neutropenia and nasopharyngeal microbiota profile. We use negative correlation for continuous variables. Page 6 line 199, this retrospective study did not assess need for prophylactic antibiotic treatment which was administered in 20% of patients. Therefore this statement should be omitted or discussed with caution Page 6, lines 201-215 make it more concise and discuss your results from the perspective of the related studies Page 6 line 216-227 discuss your results regarding febrile neutropenia Page 7 lines 228-244 check for consistency with page 4 line 152 The fact that C-rp is higher in RSV infected infants with microbiota is not mentioned in the results and appears in the discussion Page 7 lines 245-268 make this paragraph more concise as your study provides no data on cytokines and inflammatory response. You only evaluated ANC. Page7,8 lines 269-283 please include in limitations that 32% of patients did not complete follow up assessment, missing data was not performed.

Response: According to your suggestion, “transient neutropenia” is used in Discussion instead of “neutropenia”.

To explain the association between transient neutropenia and nasopharyngeal microbiota more clearly, “Median ANC andC-reactive protein level on admission were higher in the 135 infants with any of these 3 bacteria identified from nasopharyngeal aspirates (4.0 ×109/L and 8.1 mg/L) than in the 99 infants without such a microbiota profile (2.0 ×109/Land 0.9 mg/L, P <0.0001 and P <0.0001, respectively).” is added in result section. In addition, “a negative association” is changed to “an association” to prevent confusion.

As you suggested, we did not assess the need of antibiotics. “without the need for prophylactic antibiotic use” is changed to “without developing severe infection or hematologic malignancy” in the first paragraph of discussion.

According to your suggestion, in the second paragraph of discussion, we focused on the risk of severe infection and hematologic malignancy in RSV-infected with febrile neutropenia and on the diagnostic and therapeutic managements.

The consistency is checked between results and the 4th paragraph of discussion. “Median ANC and C-reactive protein level on admission were higher in the 135 infants with any of these 3 bacteria identified from nasopharyngeal aspirates (4.0 ×109/L and 8.1 mg/L) than in the 99 infants without such a microbiota profile (2.0 ×109/L and 0.9 mg/L, P <0.0001 and P <0.0001, respectively).” is described in the result section.

According to your suggestion, detailed description about cytokines and chemokines are deleted in the 5th paragraph of discussion.

Based on your suggestion, “32% of patients did not complete follow-up assessment in this study and missing data were not included in the analyses.” is added in the limitation.

Round 2

Reviewer 1 Report

This reviewer is satisfied with the changes made in this revised version of the manuscript. No further comments from me.

Reviewer 2 Report

No more comments.